# Mechanical and Gas Barrier Properties of Poly(Lactic Acid) Modified by Blending with Poly(Butylene 2,5-Furandicarboxylate): Based on Molecular Dynamics

**DOI:** 10.3390/polym15071657

**Published:** 2023-03-27

**Authors:** Ye Wang, Gongliang Jiang, Xiancheng Shao, Shikun Pu, Dengbang Jiang, Yaozhong Lan

**Affiliations:** 1School of Materials and Energy, Yunnan University, Kunming 650091, China; 2Yunnan Lincang Xinyuan Germanium Industry Co., Ltd., Lincang 677000, China; 3Green Preparation Technology of Biobased Materials National & Local Joint Engineering Research Center, Yunnan Minzu University, Kunming 650500, China

**Keywords:** polylactic acid, modification, molecular dynamics, mechanical properties, barrier properties

## Abstract

Three blends of Poly(butylene 2,5-furandicarboxylate) (PBF) and Poly(lactic acid) (PLA) blends were modeled using molecular dynamics simulations, with PBF contents of 10%, 20%, and 30%, respectively. The study investigated the compatibilities of the blends, as well as the mechanical and gas barrier properties of the composite systems. The molecular dynamics simulation results show that: (1) PLA and PBF have good compatibility in the blend system; (2) the optimal toughness modification was achieved with a 20% PBF content, resulting in a 17.3% increase in toughness compared to pure PLA; (3) the barrier properties of the blend for O_2_, CO_2,_ and N_2_ increased when increasing the PBF content. Compared to pure PLA, the diffusion coefficients of the O_2_, CO_2,_ and N_2_ of the blends with 30% PBF decreased by 75%, 122%, and 188%, respectively. Our simulation results are in good agreement with the actual experimental results.

## 1. Introduction

Bio-based polyesters are increasingly being used in various areas such as tissue engineering, food packaging engineering, and drug delivery, owing to their excellent mechanical, crystalline, and gas barrier properties [1,2,3]. Biodegradable bio-based polyesters are a class of materials with a closed loop of recycling life, where ‘resource’ and ‘waste’ are not defined [4], i.e., waste can also be transformed into resources by biological or chemical means. In contrast to petroleum-based polyesters, which are non-renewable or have a very long degradation cycle, research on renewable bio-based polyesters can effectively ensure the sustainability of polymers [5]. At present, there are still many reports on improving the recovery rate of petroleum-based polyester, but the recovery rate is not ideal. We think that this is a transitional means to achieve the goal of replacing petroleum-based polyester with bio-based polyester. A number of researchers have reported scientific methods used to increase the rate of bio-based degradation as well as effective recovery [6], which holds promise for the mass application of bio-based polyesters. It is undeniable that bio-based polyesters release large amounts of CO_2_ during degradation, which, to some extent, also contributes to the adverse effects of the greenhouse effect. In addition, reducing the cycle time and increasing the recycling rate of bio-based polyesters remain great challenges. However, from the point of view of disposal methods for waste petroleum-based polyester products, such as incineration, landfill, and dumping into the sea [7,8,9], bio-based polyesters can still provide greater ecological benefits than petroleum-based polyesters.

Polylactic acid (PLA) is one of the most promising bio-based polyesters for industrial applications. PLA can be obtained by lactic acid polycondensation or lactide ring-opening polymerization, and the end products of its degradation are carbon dioxide and water [10]. PLA has excellent mechanical properties, such as its tensile strength, flexural strength, and Young’s modulus. However, it is hard and brittle [11]. Compared to other bio-based polyesters, PLA exhibits disadvantages, such as a poor toughness and weak barrier effect [12,13]. These disadvantages have limited the development and application of PLA.

Recently, there has been considerable interest in the bio-based monomer called 2,5-furandicarboxylic acid (2,5-FDCA), which is a renewable diacid and considered as an alternative to benzodicarboxylic acid [14]. 2,5-FDCA is available from a wide range of sources and can be obtained from galactose and fructose [15,16]. The synthesis of Poly(ethylene 2,5-furandicarboxylate) (PEF) and Poly(butylene 2,5-furandicarboxylate) (PBF) via 2,5 furandicarboxylic acid has attracted the attention of researchers [17]. Among these, PBF is an emerging bio-based polyester that has received popular attention owing to its excellent crystallinity, barrier properties, and mechanical properties [18,19]. Therefore, in this study, PBF was chosen to modify the mechanical and gas barrier properties of PLA.

Computational simulations are being increasingly used in various fields, particularly in materials engineering and pharmaceutical engineering. These simulations enable us to design models and study the physical and chemical properties of materials. The use of computational simulation software in simulating reactions provides a theoretical basis for experiments, reduces costs, and increases safety. In recent years, molecular dynamics and computational simulations have been extensively used by polymer modifiers to study polymer modification, yielding meaningful theoretical predictions and scientific rationale. Fojtíková [20] investigated the dependence of the degree of cross-linking of polydimethylsiloxane (PDMS) on the bulk modulus using molecular dynamics, and the results show that the degree of cross-linking of PDMS is proportional to the bulk modulus, which is consistent with their experimental results. Rastegar [21] investigated the toughening effect of boron nitride nanosheets (BNNSs) on PLA through computational simulations based on molecular dynamics and found that the Young’s modulus and toughness of PLA-based composites increased by 24.5% and 4%, respectively, with the addition of 3 wt% BNNS. Song [22] calculated the thermal and mechanical properties and found that the glass transition temperature and modulus of the blends increased with an increasing PBI content. However, there are few reports on using molecular dynamics simulations to study the barrier properties of PLA.

In this study, we used Materials Studio (MS) software to carry out PBF modification of PLA based on molecular dynamics (MD). We analyzed the compatibility of PLA with PBF and the effect of different PBF contents on the toughness and gas barrier properties of PLA at the microscopic level. The properties of polymer materials exhibit diversity due to the complexity and multi-level nature of polymer chain structures. We used a molecular dynamics simulation to study the properties of PLA and PBF blends from a microscopic perspective, which is conducive to accelerating the research and development of PLA-based special performance materials. At the same time, studying the relationship between the polymer structure and gas permeability at the molecular level will help to understand and explore the microscopic mechanism of the polymer gas barrier in essence.

## 2. Simulation Details

In this study, all simulated processes used the COMPASSII force field. Periodic boundary conditions were applied in all directions of the polymer box. The schematic diagram of periodic boundary conditions is shown in Figure 1. Area 5 is a cubic structure with volume V, and the simulation unit contains N particles. Periodic boundary conditions are based on the assumption that there are infinitely many simulation units around area 5 that are exactly the same as it. Each unit has the same state, the same number of internal particles, and the same position and speed. When a certain number of particles leave the unit from one side of the simulation unit, the same number of particles will inevitably enter the simulation unit from the opposite side, so the particles in the simulation unit remain unchanged. After the introduction of periodic boundary conditions, the calculation of the molecular dynamics simulation only needs to calculate the original cell of the material, which greatly reduces the calculation amount of the molecular dynamics simulation. The assumption of periodic boundary conditions breaks away from the limitation of computer operation ability and avoids the finite size effect in the simulation system. In this paper, the cut-off radius was set to 12.5 Å to truncate the non-bonded van der Waals forces and electrostatic interactions. An Andersen thermostat and barostat were used to control the temperature and pressure of the simulation [23].

### 2.1. Construction of Polymer Chain Model

This study was based on molecular dynamics principles and investigated the structure and properties of PLA/PBF blends at the atomic scale, where the structural diagram of the material is shown in Figure 2. The PLA model was constructed using L-polylactic acid monomer units. It is well known that the selection of a suitable polymer chain length is a crucial step in the modeling and simulation process. Chain lengths that are too short may cause end effects to the extent that they are not representative of real material. In addition, polymer chains that are too long require a high computational power and simulation time. To facilitate this study, the molecular weights of the two polymer chains should be similar. By calculating the solubility parameters for molecular chains of different chain lengths, the minimum chain length that is representative of the polymer can be analyzed once the solubility parameters have stabilized. Based on the principles of reasonableness and molecular weight equivalence, we calculated the relationship between the number of repeat units and the solubility parameters of PLA and PBF; the results are shown in Figure 3. The solubility parameters of PLA and PBF tended to be stable when the number of repeating units was 25 and 15, respectively. PLA chains with a degree of polymerization of 61 and PBF chains with a degree of polymerization of 24 were selected for this study. Both the constructed monomer and polymer chains need to be geometrically optimized using the smart minimizer method until energy equilibrium is reached, after which the energy equilibrium chains are used for the next step of the study.

### 2.2. Construction of Polymer Model

Polymer models were constructed using energy-balanced PLA and PBF chains, with ten chains added to each model, depending on the computer capabilities. Pure PLA, PBF, and PLA/PBF blends in the ratios of 9/1, 8/2, and 7/3 were constructed as shown in Figure 4. In this study, the three blend ratios were named AF10, AF20, and AF30, where the numbers represent the percentages of PBF in the blend. The initial density of all amorphous cells was 0.8 g/cm^3^ to ensure that the polymer chains had sufficient relaxation space in the box. Too large an initial density could potentially lead to irrational structures, such as entanglement and overlap between chains. The constructed polymer model requires energy minimization, annealing, and molecular dynamics simulations to obtain an equilibrium amorphous polymer model. The energy-minimized O_2_, N_2,_ and CO_2_ were added to the polymer model according to the above steps for the study of the polymer gas barrier properties.

### 2.3. Model Optimization

First, the constructed polymer model was subjected to geometry optimization with 25,000 iterations. PLA, PBF, and three blend systems underwent 25,000 iterations of geometric optimization calculations, and their energy reached a stable state. The energy convergence curves of AF10, AF20, and AF30 are shown in Figure 5. An energy-minimized polymer model was used in the next step of the annealing simulation. In the second step, the polymer model was annealed 50 times using the Forcite module, where the temperature was increased from 298 K to 498 K and then cooled to 298 K in one cycle, with a temperature gradient of 50 K. After each cycle, the polymer model was subjected to geometry optimization and a 20 ps MD simulation. The ensemble selection for MD simulation was constant pressure and constant temperature ensemble (NPT) (P = 0.1 MPa, T = 298 K).

The model with the lowest energy after annealing was used in the next step of the MD simulation. MD simulations were carried out in the following three steps by alternating between the canonical ensemble (NVT) and NPT ensembles. First, the polymer model was simulated for 1000 ps in the NVT (T = 298 K) ensemble, releasing any unreasonable tension that may be present in the polymer. The temperature change of the blend system during the dynamic simulations of NVT ensemble is shown in Figure 6. Generally, if the temperature of the system fluctuates within ±5 K of the set temperature, the system is considered stable and can be used to study the properties of materials. Second, a further 2000 ps of MD simulation under the NPT (P = 0.1 MPa, T = 298 K) ensemble was performed to bring the density of the polymer model close to that of the real material. Finally, in order to create a zero-initial stress state and collect the trajectory files of the polymer model for analysis, an MD simulation was performed at 3000 ps under the NVT (T = 298 K) ensemble.

After the stable configuration was obtained, the compatibility between materials was first studied, because the compatibility of materials is one of the key factors that effectively improve the performance of composite materials. Good compatibility between materials can often better retain the original advantages of the two materials, so it produces good synergy. Thereafter, the mechanical properties, free volume, and diffusion coefficient of the material were studied.

We also calculated the density of PLA and PBF through molecular dynamics simulation and compared it with experimental data. As shown in Table 1, our simulation results are very close to experimental data. This further confirms that our model and calculation parameters are suitable for PLA and PBF systems.

## 3. Results and Discussion

### 3.1. Compatibility of PLA and PBF

To accurately investigate the effect of PBF on PLA and the compatibility of the two, the properties of the blends were investigated using solubility parameters δ and intermolecular pair correlation functions g(r), where intermolecular pair correlation functions are also called radial distribution functions. The solubility parameter is an important physical quantity that reflects the intermolecular forces of a material and can be used to determine the compatibility of two materials. Hildebrand put forward the concept of solubility parameters, but Hildebrand solubility parameters are only applicable to non-polar materials and are often used to predict the compatibility of the solvent and solute, where Equation (1) defines the solubility parameter [24,25]. On this basis, Hansan proposed three-component solubility parameters, which are extended to polar materials such as polymers. The relationship between Hildebrand solubility parameters and Hansan solubility parameters is shown in Equation (2) [26]. More importantly, researchers proposed replacing Hansan’s solubility parameters by calculating the bicomponent solubility parameters composed of electrostatic and van der Waals components. This method has been proven to be able to be used to study the solubility parameters of polar materials. The bicomponent solubility parameters definition is shown in Equation (3) [27,28]. In Materials Studio, the electrostatic solubility parameters and van der Waals solubility parameters can be directly obtained through the cohesive energy density calculation task in the Forcite module so as to calculate the solubility parameters of the polymer. In this study, we calculated the solubility parameters for pure PLA and pure PBF, and the results are shown in Table 1.
(1)$$\delta =\sqrt{\frac{E_{\mathrm{coh}}}{V}}=\sqrt{\mathrm{CED}}$$

In Equation (1), E_coh_ is the cohesive energy, V is the mixing volume of the material, and CED is the cohesive energy density.
(2)$$\delta^{2}=\delta_{D}^{2}+\delta_{P}^{2}+\delta_{H}^{2}$$

In Equation (2), *δ_D_*, *δ_P_*, and *δ_H_* represent the dispersion solubility parameter, polar solubility parameter, and hydrogen bonding solubility parameter, respectively.
(3)$$\delta^{2}=\delta_{elec}^{2}+\delta_{vdW}^{2}$$

In Equation (3), *δ_elec_* and *δ_vdW_* represent the solubility parameters of electrostatic and van der Waals, respectively.

According to the results in Table 2, the inter-molecular interactions of both materials were dominated by van der Waals forces.

The solubility parameters of PLA and PBF are very similar, with a ${\left(\delta_{A}-\delta_{B}\right)}^{2}$ value less than 4 J/cm^2^. Based on the principle of similar solubility, we believe that the two materials are compatible [30,31].

The radial distribution function g(r) was used to analyze the compatibility of the two materials more accurately. The radial distribution function, also known as the intermolecular pair correlation function, is the probability of finding particle A at a distance r from particle B, and g(r) can reflect the structural characteristics of the material at the microscopic level and reveal the nature of interparticle interactions in the material [32]. The expression for the radial distribution function is given by Equation (4), and the compatibility of the two materials in a blend system can be analyzed using the radial distribution function. If the interaction between A-A and B-B is less than A-B, the two materials can be considered as miscible; otherwise, the two materials are not miscible [33,34]. We analyzed the interactions between PLA and PBF molecular chains in AF10, AF20, and AF30, and the calculated g(r) values are shown in Figure 7 and Figure 8.
(4)$$g_{\mathrm{AB}}\left(r\right)=\frac{1}{4{\pi r}^{2}}\frac{1}{\rho_{A}N_{B}}\sum_{i\in B}^{N_{B}}\;\sum_{i\in A}^{N_{A}}\delta \left(r-r_{\mathrm{ij}}\right)$$

In Equation (4), N_A_ and N_B_ are the atomic numbers of materials A and B in the system, respectively, ρ_A_ is the average number density of atoms in the molecular chain of material a, r_ij_ is the distance between atoms in different molecular chains of materials A and B, and δ is the Dirac function.

The relationship between the radial distribution functions of the intramolecular C–C pairs of pure PLA and the three blends and the composition of the blends is shown in Figure 7. As shown in Figure 7, the highest characteristic peak of PLA appears at 1.55 Å, which corresponds to the C-C pairs directly bonded in the PLA molecular chain, and the subsequent peak corresponds to the C-C pairs that are not bonded. With a decrease in the PLA content in the blends, the heights of the characteristic peaks increased gradually. In the binary blend compatibility system, a strong interaction was observed between the two components, which caused the radial distribution function of C-C pairs between molecules of different components to be higher than that of C-C pairs between molecules of the same component. The calculations in Figure 8 show that the g(r)s between the molecular chains of PLA and PBF are both higher than those of PLA-PLA and PBF-PBF. This result indicates that the interaction between the molecular chains of PLA and PBF was greater than that between PLA-PLA and PBF-PBF. When PLA is mixed with PBF, the PLA chains are more likely to be in contact with the PBF chains, and the two materials are less likely to be immiscible and less likely to delaminate. The combined analysis of the solubility parameters and radial distribution function shows that the compatibility between PLA and PBF is good, and this result provides us with very valuable predictions for both computational simulations and experiments.

### 3.2. Mechanical Properties

In the Fortite module, the stiffness matrix C_ij_ of the material was calculated. The stiffness matrix of the PLA is shown in Equation (5). The Lamé constant of the material was calculated using a stiffness matrix. The Lamé constants of the PLA stiffness matrix are λ = 2.0261 and μ = 1.4711.
(5)$$\left|C_{ij}\right|=\left[\begin{array}{cccccc} \;4.5362 & \;2.0116 & \;2.1239 & -0.0488 & -0.2801 & \;0.0202\\ \;2.0116 & \;5.1619 & \;2.3172 & \;0.1217 & -0.3247 & \;0.0847\\ \;2.1239 & \;2.3172 & \;5.2068 & -0.0188 & -0.0912 & \;0.0060\\ -0.0488 & \;0.1217 & -0.0188 & \;1.6304 & \;0.0516 & -0.2268\\ -0.2801 & -0.3247 & -0.0912 & \;0.0516 & \;1.5000 & \;0.0268\\ \;0.0202 & \;0.0847 & \;0.0060 & -0.2268 & \;0.0268 & \;1.2828 \end{array}\right]$$

The tensile modulus (*E*), shear modulus (*G*), and bulk modulus (*K*) of the material can be obtained using the Lamé constants obtained using Equation (6) [35].
(6)$$E=\frac{\mu \left(3\lambda +2\mu \right)}{\lambda +\mu },G=\mu ,K=\mu +\frac{2}{3}\lambda$$

Based on the elastic static method, we obtained the tensile modulus, shear modulus, and bulk modulus of pure PLA, pure PBF, and PLA/PBF blends using the above equations, the results of which are listed in Table 3.

The modulus of a material represents its hardness and stiffness. In general, the higher the modulus of a material, the greater the hardness and stiffness of the material. As shown in Table 3, PLA is a hard and brittle material, with a much higher modulus than PBF, where the tensile modulus and shear modulus of PLA are 3.7945 and 1.4711, respectively, and the tensile modulus and shear modulus of PBF are 2.9155 and 1.0804, respectively. Therefore, the toughness of PLA can be improved by blending it with PBF. The tensile and shear moduli of the blends decreased linearly with an increasing PBF content, indicating that the addition of PBF can affect the magnitude of the tensile and shear moduli of PLA, but has little effect on the bulk modulus of PLA, with no certain pattern. The K/G value is commonly used to evaluate the toughness of a material, where a higher value indicates better toughness.

The addition of different amounts of PBF improves the toughness of PLA due to the excellent toughness of PBF, and compatibility plays a very important role in blend modification. Among the different blends, the blend with 20% PBF showed the best toughness, with a 17.3% increase in toughness compared to pure PLA at a loss of 5.3% of the tensile modulus (i.e., Young’s modulus). Long et al. [36] experimentally measured elongation at break results for the blends and showed that when the PBF content reached 20 wt%, the elongation at the break of the blends was as high as 223%, which is approximately 30 times higher than the elongation at the break of PLA. This is consistent with the mechanical property calculations in this study, where the addition of PBF effectively improved the toughness of the PLA.

### 3.3. Free Volume

The size of the free volume plays a very important role in the diffusion behavior of gas molecules in polymers. The free volume is the space wherein gas molecules can reach when diffusing. In the MS simulations, the free volume was obtained using the hard sphere probe method. The unoccupied space in the constructed box was divided by the total volume of the box to obtain the free volume fraction (FFV) in the model. In general, the larger the FFV, the greater the diffusion coefficient of the diffusing molecules in the model. We calculated the free volumes of polymers and blends through simulations in the Atom Volume & Surface tool in MS. The radii of the hard sphere probes were set to 1.52 Å, 1.53 Å, and 1.65 Å, corresponding to the kinetic radii of O_2_, N_2,_ and CO_2_, respectively [37,38]. The free volume distributions of the three gas molecules in the model are shown in Figure 9, and the FFV values are listed in Figure 10. The kinetic radii of O_2_ and N_2_ are very similar; therefore, their free volume distributions are similar. The FFV of O_2_, N_2,_ and CO_2_ tends to decrease and then increase with an increasing PBF content, but they are all smaller than the FFV in pure PLA. This indicates that the diffusion behavior of the three gases is limited in the blend, whereas it is easier to diffuse in PLA.

### 3.4. Diffusion Coefficient 

In the Forcite module, the mean square displacement (*MSD*) can be calculated via tabulated analysis [39]. The Einstein method is defined by Equation (7), which relates the diffusion coefficient D to the mean square displacement of the molecule, averaged over the number of diffusing atoms N [40]. Therefore, the equation can be simplified as D = m/6, where m is the slope of the fitted line obtained by least-squares fitting the MSD to the curve at time *t*.
(7)$$D=\frac{1}{6}\frac{d}{dt}\underset{t\underset{}{\to }\infty }{\lim}MSD\left(t\right)=\frac{1}{6}\frac{d}{dt}\underset{t\underset{}{\to }\infty }{\lim}\langle {\left|\vec{r}\left(t\right)-\vec{r}\left(0\right)\right|}^{2}\rangle$$

We calculated the *MSD* of O_2_, N_2,_ and CO_2_ in the pure polymer and co-blends and obtained the diffusion coefficients of small gas molecules in the polymer by MSD to assess the gas barrier properties of the materials using the diffusion coefficients. The results of the gas diffusion coefficient calculations for pure PLA, pure PBF, and the three blends are shown in Figure 11a. Among them, PBF has the smallest diffusion coefficient for three gases, whereas PLA has the largest diffusion coefficient. This result is consistent with the free volume result in Section 3.3, where the three gases can reach the smallest area in PBF, whereas the three gases can reach the largest area in PLA. As shown in Figure 11b, the simulation results are consistent with the free volume calculations and experimental results of Long [36]. As shown in Figure 11a, pure PBF had the best barrier properties for the three gases, which can be attributed to the polar effect of the furan ring in PBF. Owing to the good compatibility between PLA and PBF, PBF can be more uniformly dispersed in the PLA matrix. As the amount of PBF in the blend increases, the barrier properties of the blend for O_2_, N_2,_ and CO_2_ gases gradually improve, with AF30 demonstrating the best barrier properties. The results of the barrier properties of PLA, PBF, and PLA/PBF to O_2_ and CO_2_ measured by Long’s experiment [36] are shown in Figure 11b. Similar to the results of the mechanical properties, the trend of change was also consistent with our calculation results.

## 4. Conclusions

In this study, we investigated the compatibility, mechanical properties, and gas barrier properties of PLA and PBF blends using a molecular dynamics simulation. The simulation results show that PBF and PLA have good compatibility, and that the addition of PBF can significantly increase the gas barrier of PLA while improving its toughness. Specifically, adding 20% PBF improved the toughness of PLA by 17.32%, and adding 30% PBF improved the barrier properties of PLA against O_2_, CO_2_, and N_2_ by 75%, 122%, and 188%, respectively.

More importantly, by comparing with the experimental data, our simulation results are in good agreement with the actual experimental results, which shows that molecular dynamics simulation can be used as an effective tool for predicting the mechanical and barrier properties of polymer materials.

## Figures and Tables

**Figure 1 polymers-15-01657-f001:**
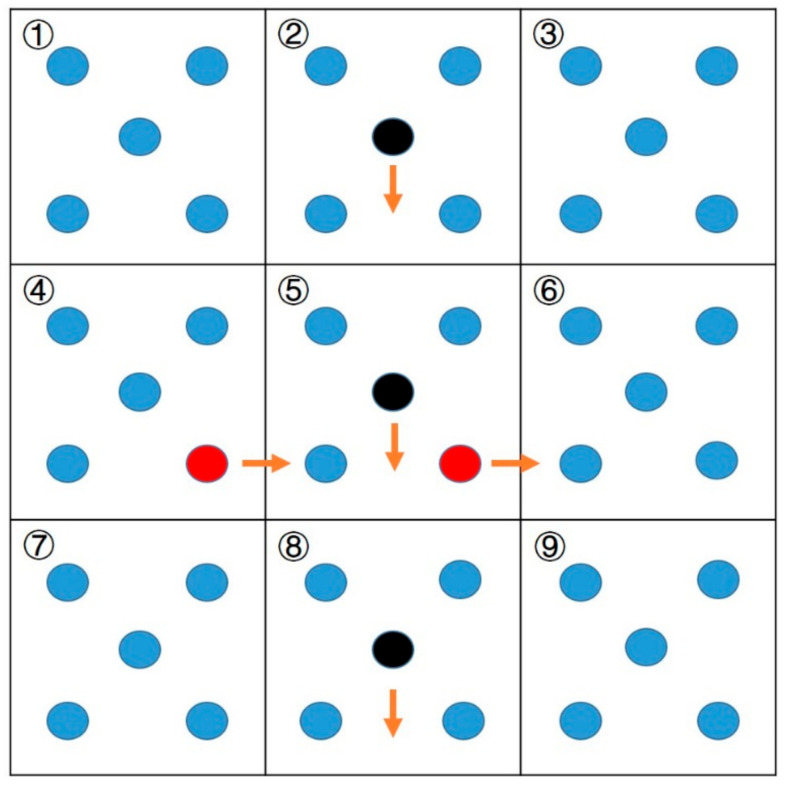
The schematic diagram of periodic boundary conditions. In the figure, the blue ball represents N particles in the box, while the black and red represent random particles in the box. The arrow represents the direction of movement of the particles, and the direction of movement is also random.

**Figure 2 polymers-15-01657-f002:**
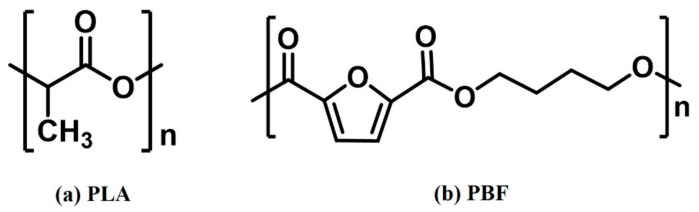
Molecular structure of (**a**) PLA and (**b**) PBF.

**Figure 3 polymers-15-01657-f003:**
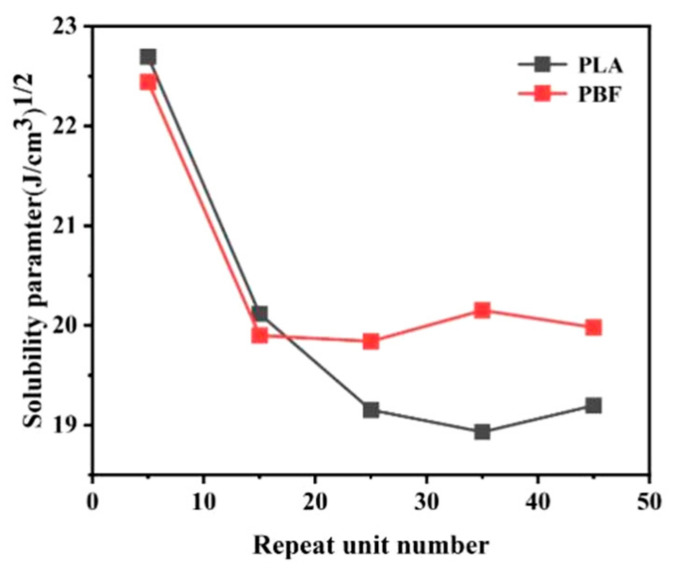
Relationship between the number of repeat units and solubility parameters of PLA and PBF.

**Figure 4 polymers-15-01657-f004:**
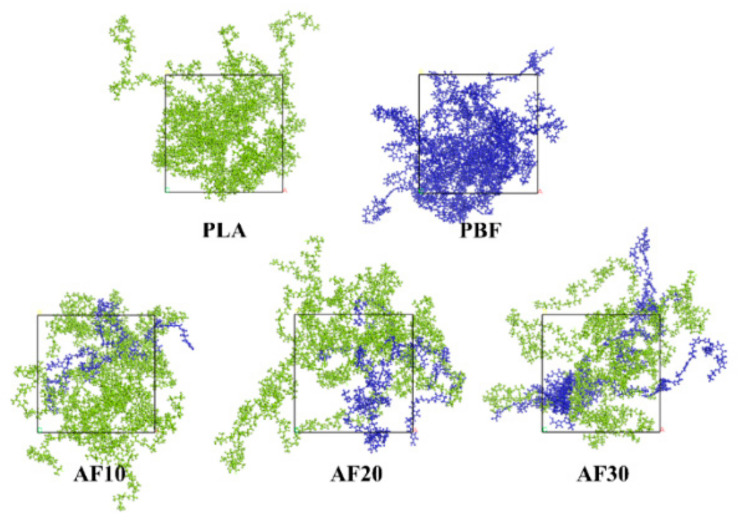
Models of pure PLA, pure PBF, and three blends. The green chain is the PLA molecular chain in the system, and the blue chain is the PBF molecular chain in the system.

**Figure 5 polymers-15-01657-f005:**
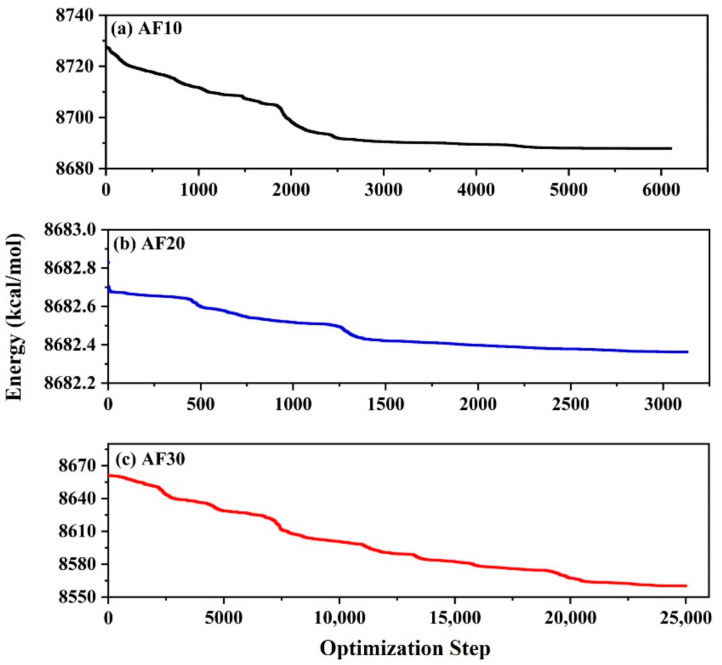
The energy convergence curves of (**a**) AF10, (**b**) AF20, and (**c**) AF30.

**Figure 6 polymers-15-01657-f006:**
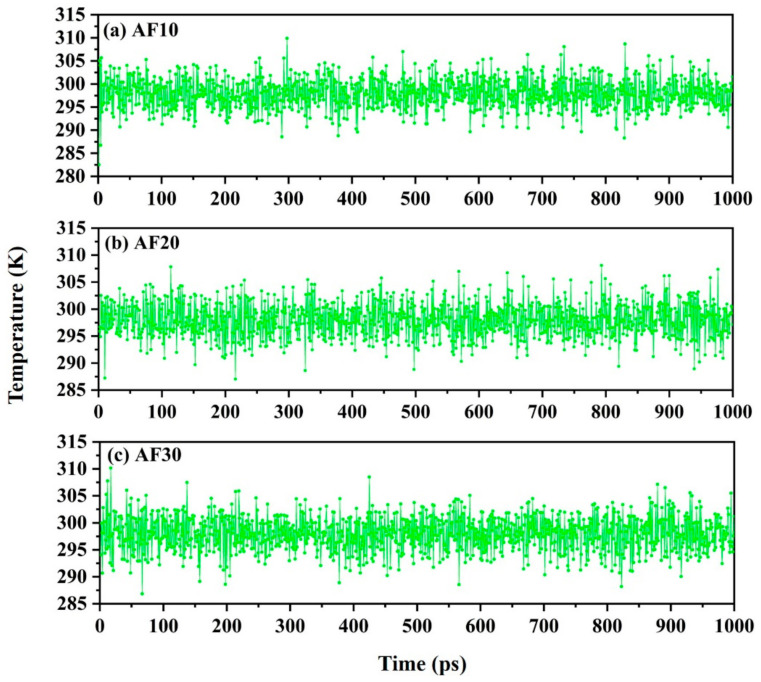
Temperature change of blend system during dynamic simulations (NVT; T = 298 K).

**Figure 7 polymers-15-01657-f007:**
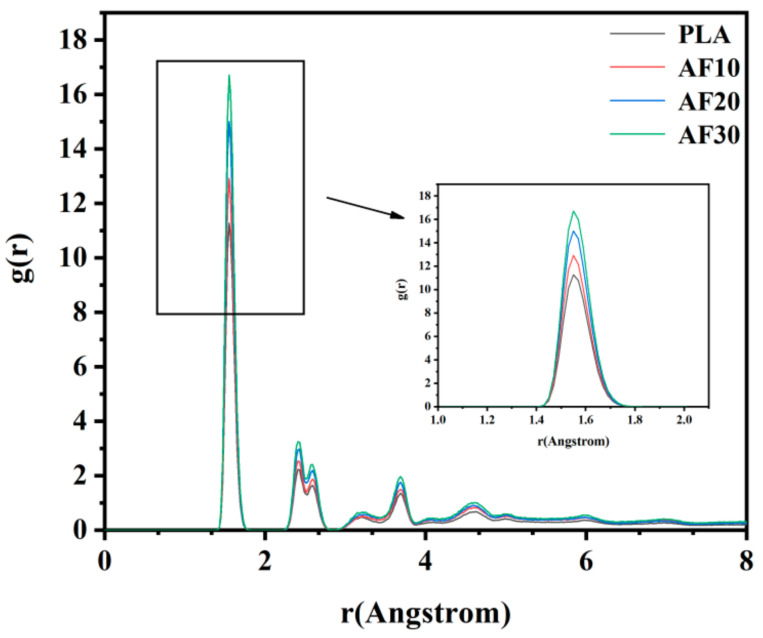
Radial distribution function of the intra-molecular carbon–carbon pairs of PLA, AF10, AF20, and AF30.

**Figure 8 polymers-15-01657-f008:**
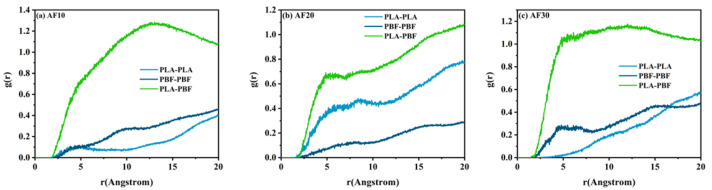
Radial distribution function of the inter-molecular carbon–carbon pairs of (**a**) AF10, (**b**) AF20, and (**c**) AF30.

**Figure 9 polymers-15-01657-f009:**
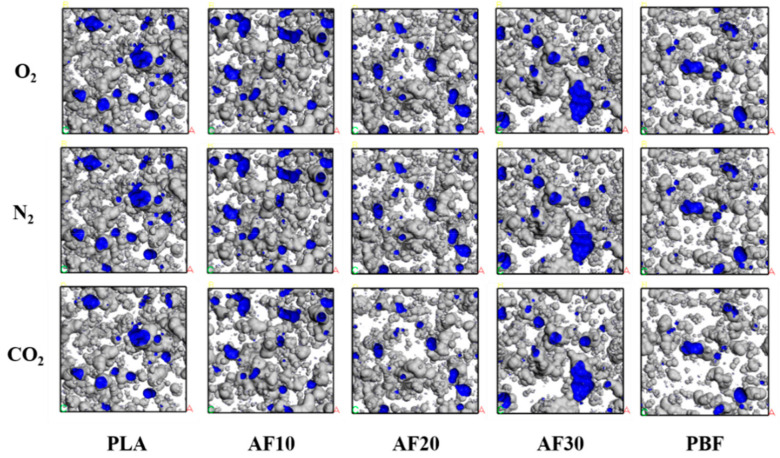
Schematic illustration of the free volume distribution of O_2_, N_2,_ and CO_2_ in PLA, PBF, and blends (3000 ps).

**Figure 10 polymers-15-01657-f010:**
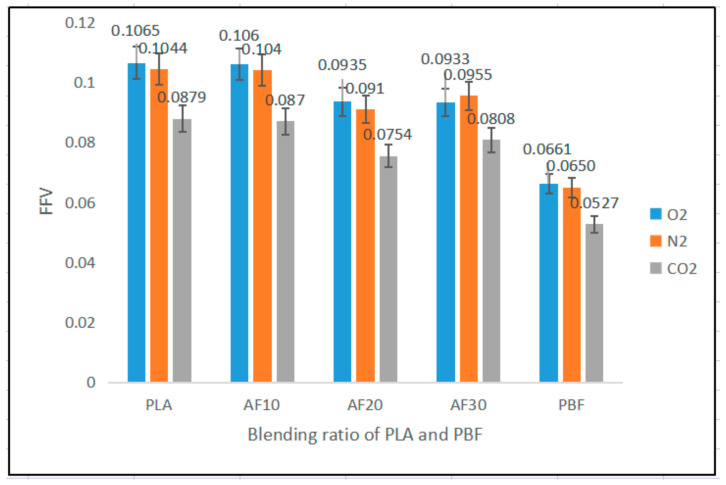
FFV of O_2_, N_2_, and CO_2_ in PLA, PBF, and blends. *p* < 0.05; error bars indicate SD; *n* = 5.

**Figure 11 polymers-15-01657-f011:**
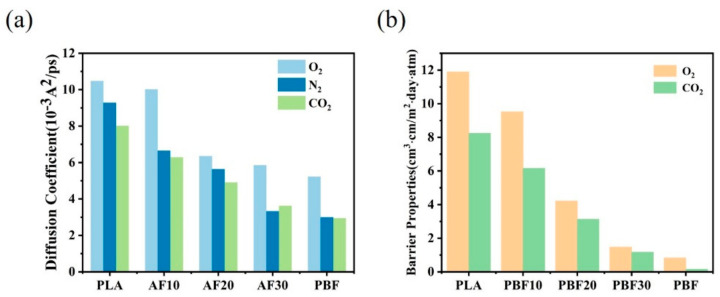
(**a**) Diffusion coefficients of O_2_, N_2,_ and CO_2_ in pure PLA, pure PBF, and blends. (**b**) Barrier properties of O_2_ and CO_2_ in pure PLA, pure PBF, and blends [36].

**Table 1 polymers-15-01657-t001:** Densities and solubility parameters of PLA and PBF from MD.

	PLA		PBF	
	Simulation	Exp. [10]	Simulation	Exp. [17]
Density(g·cm^−3^)	1.17	1.20	1.29	1.31

**Table 2 polymers-15-01657-t002:** Solubility parameters of PLA and PBF.

Polymer	δ_elec_ (J/cm^3^)^1/2^	δ_vdW_ (J/cm^3^)^1/2^	δ (J/cm^3^)^1/2^	δ (J/cm^3^)^1/2^ References
PLA	6.91 ± 0.02	16.74 ± 0.01	18.34 ± 0.01	19.28 ^a^ [29]
				19.16 ^b^ [29]
PBF	5.79 ± 0.01	18.19 ± 0.01	19.35 ± 0.01	/

^a^: Intrinsic 3D viscosity method; ^b^: intrinsic 1D viscosity method.

**Table 3 polymers-15-01657-t003:** Mechanical properties of neat PLA and neat PBF and polymer blends.

Polymer	E (GPa)	G (GPa)	K (GPa)	K/G	ε (%)
PLA	3.7945	1.4711	3.0068	2.0439	7.4 [36]
AF10	3.6998	1.4290	3.0010	2.1001	202.0 [36]
AF20	3.5944	1.3647	3.2725	2.3979	223.0 [36]
AF30	3.5167	1.3462	3.0236	2.2460	/
PBF	2.9155	1.0804	3.2237	2.9838	259.5 [36]

## Data Availability

The data presented in this study are available on request from the corresponding author.

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
