# Peer review of "Mechanical and Gas Barrier Properties of Poly(Lactic Acid) Modified by Blending with Poly(Butylene 2,5-Furandicarboxylate): Based on Molecular Dynamics"

_polymers, 2023, doi:10.3390/polym15071657_

Round 1

Reviewer 1 Report

This was a well-explained piece of work, and I think the simulations shown do lead to some reasonable conclusions. However, I am concerned that this work was done in the prior knowledge of the results obtained by Long et al (referenced in your paper), and so there is considerable risk that the parameters of the simulations were influenced consciously or unconsciously by having pre-existing “expected” outcomes. This potential source of confirmation bias should be discussed, and some effort made to demonstrate that the outcomes of the simulations are intrinsic to the system and have not been unduly influenced by prior knowledge of the experimental data. For example a study of the sensitivity to a key outcome (FFV or E) to reasonable changes in those parameters set arbitrarily by the authors (molecular weight of polymer, for example).

Minor additional comments:

-        All abbreviations (NVT, NPT etc) should be defined when first used.

-        Chemical structures should be provided for the monomers and polymers discussed to aid clarity.

-        Introduction should include some discussion of the role to be played by biodegradable materials vs recyclable materials. Biodegradable polymers like PLA are not widely recycled, and biodegradation can hamper recycling – it is worth mentioning that the desirability of biodegradability vs recyclability is application specific.

-        Is the solubility parameter discussed here the Hildebrand parameter? Please clarify, and add a brief explanation/justification of the applicability of this parameter, which is usually used for nonpolar species, to these relatively polar molecules.

-        Are FFV values stated an average of many simulations? How many? Please add an indication of the variation in these results run-to-run, perhaps an error bar or standard deviation.

Author Response

March 18, 2023

Dear Sir/Madam,

Thank you for reviewing our manuscript and offering valuable advice. In accordance with your suggestions, we have made the following revisions to our manuscript:

Point 1: This was a well-explained piece of work, and I think the simulations shown do lead to some reasonable conclusions. However, I am concerned that this work was done in the prior knowledge of the results obtained by Long et al (referenced in your paper), and so there is considerable risk that the parameters of the simulations were influenced consciously or unconsciously by having pre-existing “expected” outcomes. This potential source of confirmation bias should be discussed, and some effort made to demonstrate that the outcomes of the simulations are intrinsic to the system and have not been unduly influenced by prior knowledge of the experimental data. For example a study of the sensitivity to a key outcome (FFV or E) to reasonable changes in those parameters set arbitrarily by the authors (molecular weight of polymer, for example).

Response : This is a good question. The real molecular weight of polymer materials is very large, and due to technical limitations, it is not possible to establish real chain length polymers in simulation. The key to achieving MD simulation is to find a critical polymer chain length that does not require too much computation and can represent the properties of real polymer chains. Some studies have shown that when the number of repeating units of a polymer reaches a certain critical value, its solubility parameter( δ) Will tend to stabilize without significant changes, so calculations are used δ Determining the appropriate polymer chain length based on the relationship between the degree of polymerization can build a more accurate model for predicting the various properties of polymer materials.(Polymer,2009,50(20):4 973–4 978ï¼›J Phys Chem B,2005,109(32):15611–15620ï¼› Polymer,2006,47(23):8 061–8 071.) Before the mixed task simulation, we established PLA and PBF molecular chains with different polymerization degrees and performed MD simulation δ Based on the relationship between the degree of polymerization and the degree of polymerization, a model was constructed. The establishment of the model is not based on the work of Long et al. Although the ultimate purpose of the simulation calculation in this article is indeed to explain the work of Long et al., our model is mainly based on solubility parameters, and the solubility parameters and density of the constructed model are very close to the experimental results.This further confirms that our model and calculation parameters are suitable for PLA and PBF systems.

Point 2: All abbreviations (NVT, NPT etc) should be defined when first used.

Response: Thank you for the error you pointed out. This error was caused by our carelessness, we have corrected this error and checked for similar errors, which have been corrected in lines 81, 82, 165 and 172 respectively.

Point 3: Chemical structures should be provided for the monomers and polymers discussed to aid clarity.

Response 3: Thank you for your suggestions. The structural unit diagrams of PLA and PBF have been added to line 132, i.e., Figure 2.

Point 4: Introduction should include some discussion of the role to be played by biodegradable materials vs recyclable materials. Biodegradable polymers like PLA are not widely recycled, and biodegradation can hamper recycling – it is worth mentioning that the desirability of biodegradability vs recyclability is application specific.

Response 4: Thank you for your suggestions. At present, the waste rate of bio-based polyester products is still faster than its degradation rate. While reducing the degradation cycle of bio-based polyester, improving the recovery rate of bio-based polyester can produce synergy and relieve the pressure of waste polyester products on the ecological environment. For the discussion between biodegradability and recyclability, we have added lines 30 to 40.

Point 5: Is the solubility parameter discussed here the Hildebrand parameter? Please clarify, and add a brief explanation/justification of the applicability of this parameter, which is usually used for nonpolar species, to these relatively polar molecules.

Response : Thank you for your question. We have not explained the solubility parameters in detail in this paper. The solubility parameter in the paper is not Hildebrand solubility parameters, but the two-component solubility parameters proposed by Gupta et al., which can be used to replace Hansen solubility parameters [1-4]. The source and definition of solubility parameters have been added to lines 202 to 231 in the paper.

Point 6: Are FFV values stated an average of many simulations? How many? Please add an indication of the variation in these results run-to-run, perhaps an error bar or standard deviation.

Response : The FFV value in this paper is obtained by analyzing the last frame of molecular dynamics simulation data using the colly surface method. According to your suggestion, we conducted five FFV calculations using 2200ps, 2400ps, 2600ps, 2800ps, and 3000ps data. The data in figure 10 were expressed as mean ± standard deviation (SD). Statistical Product Service Solutions software (SPSS) was used for statistical analysis of the relevant data.

Reviewer 2 Report

Dear,

The authors carried out an investigation of PLA/PBF blends using molecular dynamics. It is a good investigation to show the behavior of polymer blends. However, I recommend developing the experimental and later the simulation to consistently validate.

> Authors must make clear the novelty of the manuscript and the contribution of the work;

> Inform the reference of the data presented in Table 1;

> I recommend detailing the validation of the model used with the constraints;

> Page 4. “Based on the principle of similar solubility, we believe that the two materials are compatible”. Why did the authors not perform an experimental dynamic-mechanical thermal analysis test to validate the statement?

Author Response

March 18, 2023

Dear Sir/Madam,

Thank you for reviewing our manuscript and offering valuable advice. In accordance with your suggestions, we have made the following revisions to our manuscript:

Point 1: The authors carried out an investigation of PLA/PBF blends using molecular dynamics. It is a good investigation to show the behavior of polymer blends. However, I recommend developing the experimental and later the simulation to consistently validate.

Response: Dear expert, your suggestions have been very helpful to us. I know that supplementary validation experiments will improve the overall level of this article and the credibility of the calculation results. However, we were unable to supplement the relevant experiments for the following reasons:There are currently no PBFs available for market sale, and we have not purchased PBFs for experiments through various channels. We have described the problem you raised as a deficiency of the paper in the discussion of the solubility parameter calculation section.

In order to verify the accuracy of our calculation data, we cited the experimental work data of Long et al. in this regard for verification.At the same time, we also calculated the density of PLA and PBF through molecular dynamics simulation, and compared them with experimental data. As shown in Table 1, our simulation results are very close to experimental data. This further confirms that our model and calculation parameters are suitable for PLA and PBF systems.

In this thesis, we mainly focus on using molecular dynamics simulation to study the compatibility, mechanical properties, and gas barrier properties of PLLA and PBF blends from a microscopic perspective. The purpose of this study is to understand and explore the microscopic mechanism of polymer gas barriers by studying the relationship between polymer structure, mechanical properties, and permeability at the molecular level. To provide a new idea for accelerating the research and development of PLLA based special performance materials.

Point 2: Authors must make clear the novelty of the manuscript and the contribution of the work.

Response: Thank you for your suggestions.We have rewritten the last paragraph of the introduction section to clarify the novelty and contribution of the manuscript.

Point 3: Inform the reference of the data presented in Table 1.

Response: Thank you for your suggestions. We have looked up some literature on the solubility parameters of PLA and PBF, but we haven't looked up the solubility parameters of PBF for the time being. Regarding the solubility parameters of PLA, we found that the solubility parameters measured by different methods are different, and the data are shown in Table 1 of this document, the solubility parameters we calculated are lower than those in the literature. In MS simulation, the solubility parameters calculated under the same limiting conditions have certain reference significance.

Table 1. Solubility parameters of PLA in references

Method

δD

(J/cm3)1/2

δP

(J/cm3)1/2

δH

(J/cm3)1/2

δ

(J/cm3)1/2

References

Intrinsic 3D viscosity method

17.61

5.30

5.80

19.28

[1]

Classical 3D geometric method

16.85

9.00

4.05

19.53

[1]

Van Krevelen’s method

15.33

8.44

10.98

20.66

[2]

Hoy’s method

14.02

12.73

9.77

21.31

[2]

the group contribution method

/

/

/

21.9

[3]

References:

  1. Agrawal A, Saran A D, Rath S S, et al. Constrained nonlinear optimization for solubility parameters of poly(lactic acid) and poly(glycolic acid) - validation and comparison [J]. Polymer, 2004, 45(25): 8603-8612.
  2. Su S. Prediction of the Miscibility of PBAT/PLA Blends [J]. Polymers, 2021, 13(14): 10.
  3. Dil E J, Carreau P J, Favis B D. Morphology, miscibility and continuity development in poly(lactic acid)/poly(butylene adipate-co-terephthalate) blends [J]. Polymer, 2015, 68: 202-212.

Point 4: I recommend detailing the validation of the model used with the constraints.

Response: Thank you for your suggestions. With regard to the validation of the model used, we have added lines 164 to 166 and 181 to 184 in the article, and added the data of the model optimization process, as shown in Figure 5 and Figure 6 in the article.

Point 5: Page 4. “Based on the principle of similar solubility, we believe that the two materials are compatible”. Why did the authors not perform an experimental dynamic-mechanical thermal analysis test to validate the statement?

Response: Dear expert, your suggestions have been very helpful to us. I know that supplementary validation experiments will improve the overall level of this article and the credibility of the calculation results. However, there are currently no PBFs available for market sale, and we have not purchased PBFs for experiments through various channels. We have described the problem you raised as a deficiency of the paper in the discussion of the solubility parameter calculation section. According to the experimental results of Long et al., adding PBF to PLLA can significantly improve the toughness of PLLA, which indirectly proves that PLLA and PBF blends do not undergo phase separation (There is good compatibility between PLLA and PBF)

Round 2

Reviewer 2 Report

The authors satisfactorily answered the questions. Therefore, the manuscript is a good contribution to the literature.